# Modeling intra-mosquito dynamics of Zika virus and its dose-dependence confirms the low epidemic potential of *Aedes albopictus*

Sebastian Lequime[1,2], Jean-Sébastien Dehecq[3], Séverine Matheus[4,5], Franck de Laval[6,7], Lionel Almeras[8,9,10], Sébastien Briolant[8,9,10], Albin Fontaine[8,9,10]*

1 Cluster of Microbial Ecology, Groningen Institute for Evolutionary Life Sciences, University of Groningen, Groningen, The Netherlands, 2 KU Leuven Department of Microbiology, Immunology and Transplantation, Rega Institute, Laboratory of Clinical and Epidemiological Virology, Leuven, Belgium, 3 French Ministry of Health, Agence Régionale de Santé de La Réunion, Vector control Unit, La Reunion Island, Saint-Denis, France, 4 Laboratory of Virology, National Reference Center for Arboviruses, Institut Pasteur, Guyane Française, Cayenne, France, 5 Environment and infections risks unit, Institut Pasteur, Paris, France, 6 SSA, Service de Santé des Armées, CESPA, Centre d'épidémiologie et de santé publique des armées, Marseille, France, 7 Aix Marseille Univ, INSERM, IRD, SESSTIM, Sciences Economiques & Sociales de la Santé & Traitement de l'Information Médicale, Marseille, France, 8 Unité Parasitologie et Entomologie, Département Microbiologie et maladies infectieuses, Institut de Recherche Biomédicale des Armées (IRBA), Marseille, France, 9 Aix Marseille Univ, IRD, SSA, AP-HM, UMR Vecteurs–Infections Tropicales et Méditerranéennes (VITROME), Marseille, France, 10 IHU Méditerranée Infection, Marseille, France

* albinfont@gmail.com

**Data Availability Statement:** All relevant data are within the manuscript and its Supporting Information files.

## Abstract

Originating from African forests, Zika virus (ZIKV) has now emerged worldwide in urbanized areas, mainly transmitted by *Aedes aegypti* mosquitoes. Although *Aedes albopictus* can transmit ZIKV experimentally and was suspected to be a ZIKV vector in Central Africa, the potential of this species to sustain virus transmission was yet to be uncovered until the end of 2019, when several autochthonous transmissions of the virus vectored by *Ae. albopictus* occurred in France. Aside from these few locally acquired ZIKV infections, most territories colonized by *Ae. albopictus* have been spared so far. The risk level of ZIKV emergence in these areas remains however an open question. To assess *Ae. albopictus*' vector potential for ZIKV and identify key virus outbreak predictors, we built a complete framework using the complementary combination of (i) dose-dependent experimental *Ae. albopictus* exposure to ZIKV followed by time-dependent assessment of infection and systemic infection rates, (ii) modeling of intra-human ZIKV viremia dynamics, and (iii) *in silico* epidemiological simulations using an Agent-Based Model. The highest risk of transmission occurred during the pre-symptomatic stage of the disease, at the peak of viremia. At this dose, mosquito infection probability was estimated to be 20%, and 21 days were required to reach the median systemic infection rates. Mosquito population origin, either temperate or tropical, had no impact on infection rates or intra-host virus dynamic. Despite these unfavorable characteristics for transmission, *Ae. albopictus* was still able to trigger and yield large outbreaks in a simulated environment in the presence of sufficiently high mosquito biting rates. Our results reveal a low but existing epidemic potential of *Ae. albopictus* for ZIKV, that might explain the absence of large scale ZIKV epidemics so far in territories occupied only by *Ae. albopictus*.

**Funding:** This study was funded by the Direction Générale de l'Armement (https://www.defense.gouv.fr/dga, grant no PDH-2-NRBC-2-B-2113, SB) and the Direction Centrale du Service de Santé des Armées (https://www.defense.gouv.fr/sante, grant agreement 2016RC10, SB) and was supported by the European Virus Archive goes Global (EVAg, https://www.european-virus-archive.com) project that has received funding from the European Union's Horizon 2020 research and innovation program under grant agreement No 653316. The contents of this publication are the sole responsibility of the authors. SL was funded by a postdoctoral grant of the Fonds Wetenschappelijk Onderzoek – Vlaanderen (FWO, https://www.fwo.be). The funders had no role in study design, data collection and analysis, decision to publish, or preparation of the manuscript

**Competing interests:** The authors have declared that no competing interests exist.

They nevertheless support active surveillance and eradication programs in these territories to maintain the risk of emergence to a low level.

## Author summary

Zika virus (ZIKV) has emerged worldwide and triggered large outbreaks in human populations. While the yellow fever mosquito *Aedes aegypti* is considered the primary vector of ZIKV, the Asian tiger mosquito *Aedes albopictus* has been shown experimentally to transmit the virus and has been involved in a few autochthonous transmission in France in 2019. Here, we provide a comprehensive study on the ability of *Ae. albopictus* mosquitoes to transmit ZIKV by considering the within-host dynamics of ZIKV infection in humans and its impact on both mosquito infection probability and time to mosquito infectiousness. These empirical data were then leveraged by *in silico* simulations to embed them into their epidemiological context. Our study reveals a low but existing epidemic potential of *Ae. albopictus* for ZIKV, whatever their tropical or temperate origins. We identified mosquito density as a predictor for ZIKV outbreak occurrence when vectored by *Ae. albopictus*. Our findings help to explain the absence of large scale ZIKV epidemics in territories occupied by *Ae. albopictus* but call for active surveillance and eradication programs to maintain the risk of emergence to a low level.

## Introduction

Several mosquito-borne viruses have spilled from their primary enzootic cycles in tropical primary forests to emerge worldwide in transmission cycles involving humans and mosquitoes highly adapted to urban environments. While the yellow fever mosquito *Aedes aegypti* is considered as the primary vector of viruses affecting human health, the Asian tiger mosquito *Aedes albopictus* might progressively take the lead because of its outstanding invasive capacity [1] and its ability to transmit numerous virus species experimentally [2]. Originating from South-East Asia, *Ae. albopictus* has indeed invaded the world and is now present in all inhabited continents, including temperate Europe, owing to its potential to endure harsh winter conditions [3,4]. The species is also displacing *Ae. aegypti* populations in areas where both co-exist due to competitive advantages, notably at the larval stage [5–7].

Zika virus (ZIKV) is the lastest globally-emerging mosquito-borne virus that can be maintained in urban cycles that only involve human-mosquito transmissions. ZIKV is an RNA virus from the *Flavivirus* genus (Flaviviridae), which includes other human pathogens such as yellow fever virus (YFV) or dengue viruses (DENV). First isolated in 1947 from a rhesus monkey in the Zika forest in Uganda, ZIKV has concomitantly emerged in 2007 in Gabon [8] and in the Federated States of Micronesia where it triggered a significant outbreak on Yap Island [9]. The virus then spread through the South Pacific islands from 2013 to 2014 [10,11] before it emerged in north-eastern Brazil in 2015 [12], where it started a large outbreak hitting a total of 50 territories and countries in the Americas [13]. Most ZIKV infections are asymptomatic or cause self-limiting febrile sickness but are sometimes associated with complications, including Guillain-Barré syndrome or congenital microcephaly or fetal losses in women infected during pregnancy [14].

*Ae. albopictus* mosquitoes are known to be competent vectors of several major arthropod-borne viruses (arboviruses) [2]. They have been incriminated in the transmission of

chikungunya virus (CHIKV) and DENV viruses in recent outbreaks in La Réunion island, a French oversea territory in the Indian Ocean [15,16], as well as autochthonous transmission cases or outbreaks in Europe [17–20]. European French populations of *Ae. albopictus* mosquitoes were shown experimentally to transmit DENV and CHIKV as efficiently as the typical tropical vector *Ae. aegypti* [21]. The presence of barriers preventing systemic ZIKV infection (virus dissemination from the midgut to secondary tissues) or transmission were repeatedly reported for *Ae. albopictus* [22–26] and suggested a limited risk of Zika virus transmission in Europe or La Réunion island [22,24]. However, *Ae. albopictus* was strongly suspected to be the main vector of ZIKV in Central Africa [8], and in three cases of autochthonous ZIKV transmissions without evidence of sexual transmission that occurred at the end of 2019 in South Metropolitan France where *Ae. albopictus*, but not *Ae. aegypti*, is well established [27]. Temperate regions invaded by *Ae. albopictus* are of high risk of importation of mosquito-borne viruses. Indeed, because of extensive trade and tourism, these are highly connected to regions repeatedly hit by mosquito-borne viruses.

The intrinsic ability of a mosquito to be infected and subsequently transmit a virus, termed vector competence, is strongly influenced by numerous factors, including the infectious oral dose received by mosquitoes and the time post-exposure [28]. In the present study, we assessed the susceptibility of *Ae. albopictus* populations from Marseille (Metropolitan France) and La Réunion island (French overseas territory in the Indian Ocean) for ZIKV infection and compared their intra-host systemic infection dynamics. We also experimentally investigated how intra-human ZIKV load dynamics influence mosquito infection probability and systemic infection dynamics. We then implemented *in silico* epidemiological simulations to leverage these empirically acquired data into a realistic framework to assess the potential of *Ae. albopictus* as a vector of ZIKV and to identify strong virus outbreak predictors.

## Results

### Modeling of ZIKV loads dynamic in human venous blood

We aimed to provide a realistic Zika viremia dynamic in human post-infection based on viral loads in sera collected from human venous blood. ZIKV loads in venous sera from symptomatic patients ranged from 0 to $3.2 \times 10^5$ RNA copies/mL in the post-symptomatic stage of the infection. One serum had an estimated virus load of $5 \times 10^5$ RNA copies/mL on the day before symptom onsets. ZIKV load dropped by 2 log points (*i.e.*, $5 \times 10^3$ RNA copies/mL) for this patient one day after symptom onsets. We fitted a target-cell limited model [29] with a latent phase on viral genomic loads dynamics averaged across patients using parameter values estimates from experimental assays in non-human primates [29] as strong informative priors. The numerical integration of the model is plotted in Fig 1A (gold line) with ZIKV loads from human venous blood as dots. Prior distributions had a considerable influence on the posterior distribution for parameters *T0* (initial uninfected target cells concentration, 45% prior posterior overlap (PPO)), *V0* (initial concentration of free virus particles, 43% PPO), *k* (rate of cell transition from a non-productive sate to a virus productive state, 35% PPO) and *p* (free virus releasing rate by productively infected cells, 45% PPO). Our data were not informative enough to overcome the influence of the prior distribution for these parameters (S1 Fig).

Our modeled ZIKV plasma loads dynamic in humans was characterized by a period of exponential growth that reaches a peak of $5.4 \times 10^7$ ZIKV RNA copies/mL at day 2 post-infection (equivalent to an estimated $7.2 \times 10^4$ FFU/mL), followed by a period of exponential decline until the virus becomes undetectable around 12 days post-infection (Fig 1A). An infectious inoculum with an estimated mean of 10,415 ZIKV RNA copies per mL (posterior sample quantiles (psq) 2.5%: 10,015; 97.5%: 11,184 RNA copies/mL) in contact with an estimated

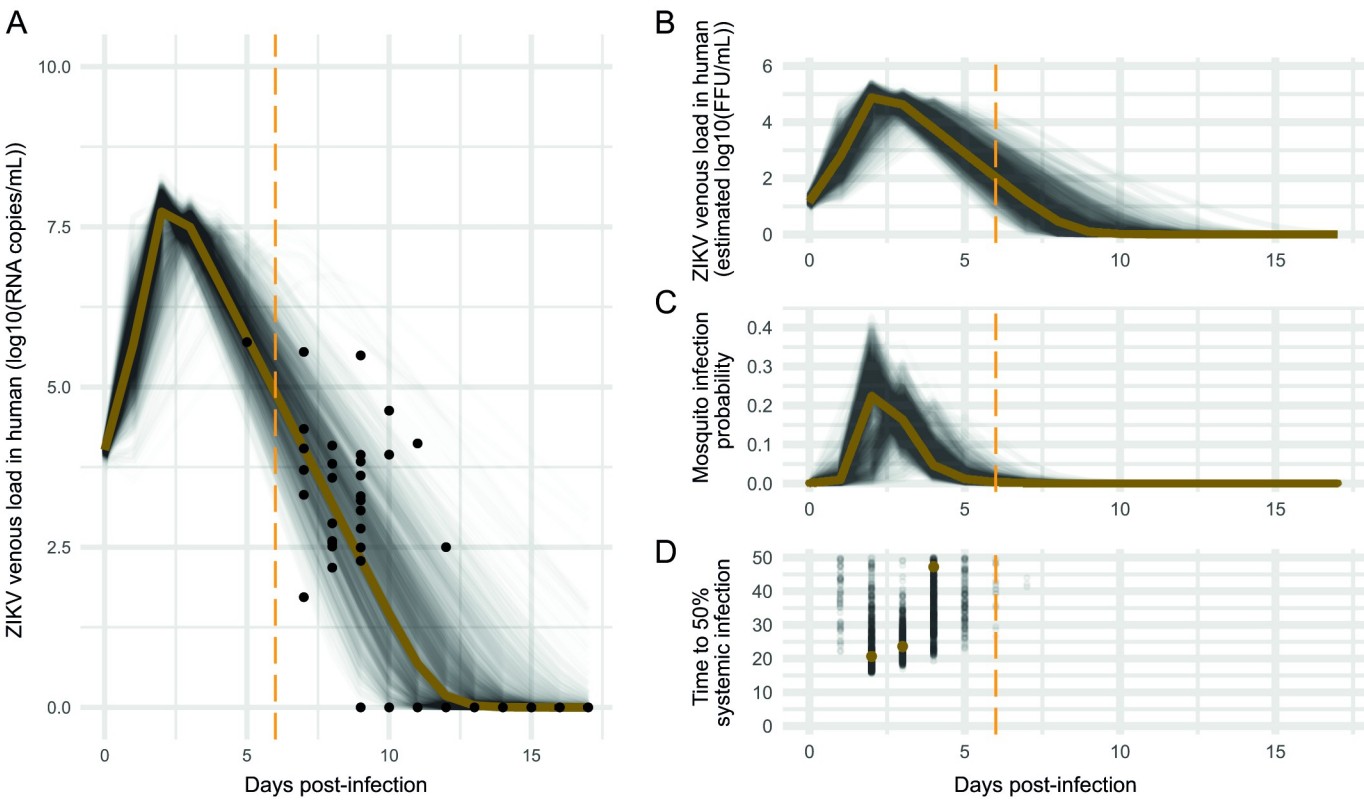

**Fig 1. Intra-human ZIKV loads dynamic modelisation and its direct impact on mosquito infection and systemic infection dynamic.** A- Estimated ZIKV viremia (RNA copies/mL) dynamic post-infection in humans. Bayesian inference was used to fit a target-cell limited model with a latent phase on viral genomic loads dynamics averaged across patients using parameter values estimates from non-human primates as strong informative priors. The golden line represents model prediction using mean fit parameter values. ZIKV loads in patients are represented with dots. Thin black lines represent model prediction for 1,000 independent individuals. For each individual, each parameter value was drawn from a normal distribution in the form $N(\mu = $ mean parameter estimate, $\sigma2 = \mu/6)$, with 6 being a scaling factor. The scaling factor was determined arbitrarily so that the distribution of predicted viremia dynamics cover patient data. B- Modelisation of ZIKV venous load (FFU/mL) dynamic post-infection in humans. The golden line represents model prediction using mean fit parameter values, as determined in A. Thin black lines represent model prediction for 1,000 independent individuals. These viremia dynamics were directly inferred from RNA copies/mL displayed in (A) by applying a conversion factor. C- Mosquito infection probability dynamic directly inferred from the intra-human ZIKV viremia in (B). Logistic regression was used to model and predict the virus dose effect on the probability of mosquito infection based on our experimental dose-response data. D- Median time to systemic infection dynamic as a function of the intra-human ZIKV viremia in (B). Median times to reach 50% systemic infection (virus dissemination) were estimated for non-null mosquito infection probabilities. The Y-axis was limited to 50 days, a time that was estimated to largely exceed the mosquito life expectancy in nature. The vertical orange dashed line represents the time to symptom onset in humans.

mean of $1.1 \times 10^5$ (psq 2.5%: $1 \times 10^5$, 97.5%: $1.2 \times 10^5$) uninfected target cells per mL initiated the ZIKV infection. The average transition time estimate from the latent phase to the virus productive phase was 2.3 hours (h) (psq 2.5%: 1.5 h; 97.5%: 3.8 h). Productively infected cells release 26,579 (psq 2.5%: 25,053; 97.5%: 29,901) free virus per cell per day and have an average lifespan of 12h. The number of secondary cells infected by ZIKV production from one infected cell in a population of target cells was estimated to 7.6.

## Infection rates and systemic infection dynamics across mosquito populations

We aimed to expose *Ae. albopictus* mosquito populations orally to ZIKV at a titre of $2 \times 10^6$ FFU/mL. This ZIKV load has a 2 log point drop from the highest ZIKV loads in human venous blood (expressed in ZIKV RNA copies per mL) that were inferred from our intra-human dynamics analysis. These virus titers also lie between median infection doses that were

reported for *Ae. albopictus* (*i.e.*, from 6.1 to 6.8 $\log_{10}$ FFU/mL) [25,26]. A total of 227 and 175 engorged female mosquitoes from Marseille and La Réunion populations, respectively, were experimentally exposed to one ZIKV isolate. The virus was isolated from the plasma of a traveler returning from Suriname in 2016 that belongs to the Asian lineage. Two ZIKV experimental exposure assays replicates were performed with infectious blood meals titered at $7.5 \times 10^6$ FFU/mL (98 and 120 engorged mosquitoes for Marseille and La Réunion populations, respectively) and $3 \times 10^6$ FFU/mL (129 and 55 engorged mosquitoes for Marseille and La Réunion populations, respectively). Because each assay was performed with a single dose, the virus dose effect was here confounded with the experiment effect, but infection and systemic infection rates can be directly compared between mosquito populations in each replicate. High body infection prevalences were obtained for both populations, independently of the time post-exposure (Fig 2A and 2B). Averaged over time, mean body infection prevalences were 84% for the Marseille population for each replicate and over 86% for the La Réunion population. A full factorial logistic regression model revealed no statistically significant difference for body infection prevalences across mosquito populations and replicates. However, there was a statistically significant interaction effect between time post-virus exposure and the replicate (analysis of deviance, p-value = 0.048), which was driven by a relatively low body prevalence at 5 days post-virus exposure for mosquitoes from the Marseille population at $7.5 \times 10^6$ FFU/mL.

Systemic infection prevalences were measured by counting the number of mosquitoes with infected heads over the number of mosquitoes with an infected body (Fig 2C and 2D). Systemic infection prevalences were significantly influenced by the time post-exposure (analysis of deviance, p-value $< 2 \times 10^{-16}$) but not by mosquito population or replicate. The intra-host dynamic of systemic ZIKV infection was inferred in all combinations of mosquito population and virus doses by fitting a logistic model. The model failed to provide relevant estimates for the La Réunion population at the lowest virus dose due to a lack of samples at early times post-virus exposure. However, systemic infection prevalences saturated at 100% for both mosquito populations and replicates. The estimated time required to reach 50% of systemic infections was 10.1 (Standard Error: 0.6) and 10 (SE: 0.8) days for the Marseille and the La Réunion populations, respectively, independently of the replicate.

## Mosquito infection rates as a function of the oral ZIKV dose

*Ae. albopictus* mosquitoes from the La Réunion population were orally challenged with two additional doses of ZIKV ($8 \times 10^5$ FFU/mL and $2.37 \times 10^8$ FFU/mL) in order to assess the effect of the virus dose on mosquito infection probabilities. Mosquitoes were held at a standardized 28°C temperature during the rearing and experimental procedure. This dose-response analysis was performed independently of the mosquito populations (*i.e.*, both populations were considered as a single one) because no difference was previously revealed for both infection and systemic infection dynamics across populations. Mosquito infection rates increased as a function of the oral ZIKV dose (S2 Fig) without a significant effect of the time post-infection (analysis of deviance). Independently of the time post-infection, the observed prevalence of mosquito infection were 57.3%, 84.5%, 87.3% and 98% for the $8 \times 10^5$ FFU/mL, $3 \times 10^6$ FFU/mL, $7.5 \times 10^6$ FFU/mL and $2.37 \times 10^8$ FFU/mL infectious blood meal concentrations, respectively. A logistic model was fit to the data by considering the blood meal virus titer as a unique explanatory variable. The infectious dose required to infect 50% (ID50) of *Ae. albopictus* mosquitoes was $4.2 \times 10^5$ FFU/mL (95% CI: $1.9 \times 10^5$ FFU/mL—$6.5 \times 10^5$ FFU/mL) (Fig 3). One-quarter of the mosquitoes were infected at $9 \times 10^4$ FFU/mL, and 95% of mosquitoes were infected at $2 \times 10^6$ FFU/mL.

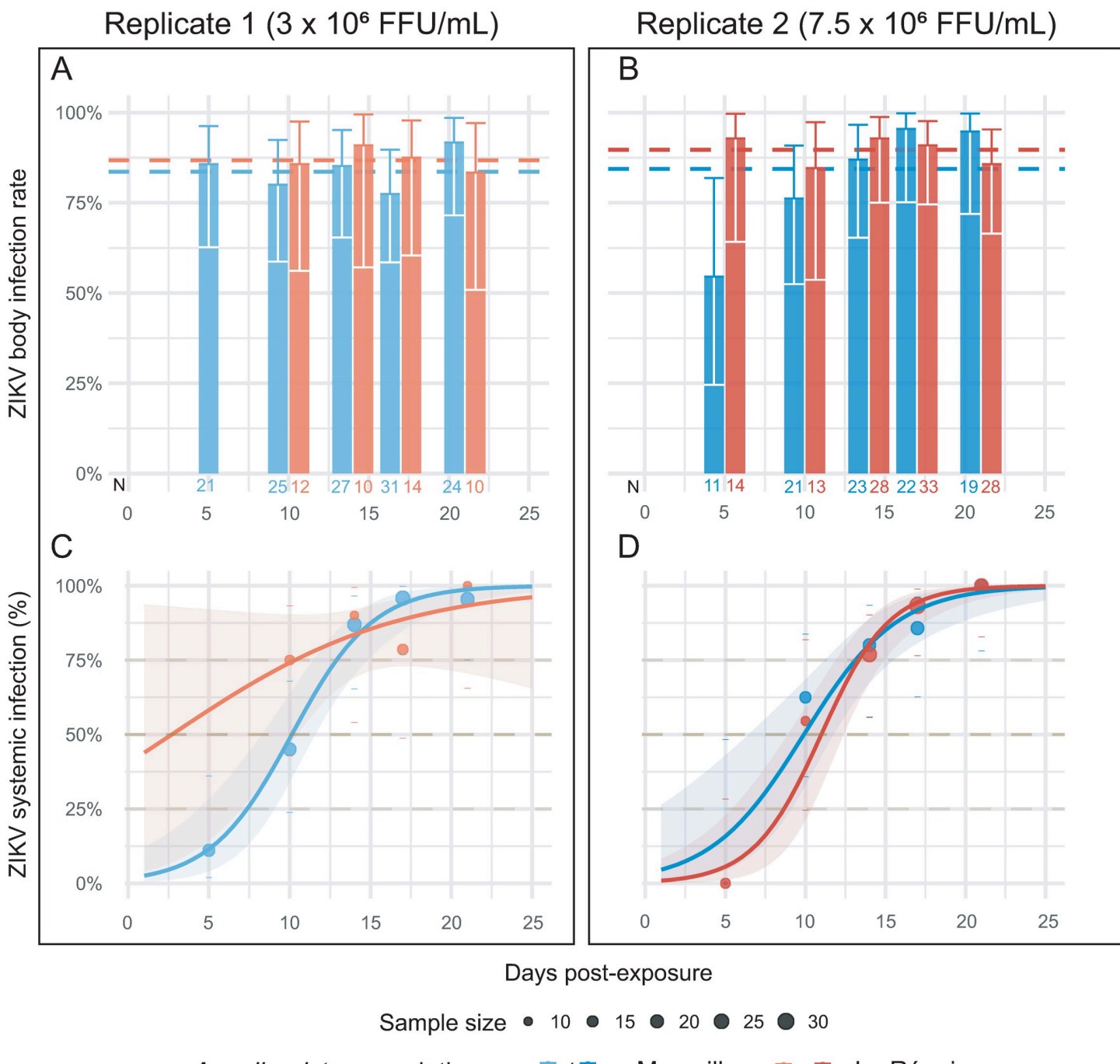

**Fig 2. Body infection prevalence (A, B) and systemic infection dynamics (C, D) for two *Ae. albopictus* mosquito populations exposed the same isolate of ZIKV.** Panel A, C and B, D represent two experiment duplicates from which the infectious blood meal was tittered at $3 \times 10^6$ FFU/mL (A, C) and $7.5 \times 10^6$ FFU/mL (B, D). Percentages of body infections over time post-ZIKV exposure are represented for the Marseille (Metropolitan France) population (blue color) and La Réunion (Overseas France) population (red color) in both replicates. The sample size is indicated below each bar. Prevalences averaged across time points are represented with a dashed line for each mosquito population. 95% confidence intervals are indicated with the error bars. Cumulative prevalence of systemic (disseminated) infections over time post ZIKV exposure are represented as points for the Metropolitan France population (blue color) and Overseas France population (red color) (C, D). Dashes represent the 95% confidence intervals of prevalences. Dots size indicates the number of samples. The fitted values obtained with a logistic model are represented for each population by a line.

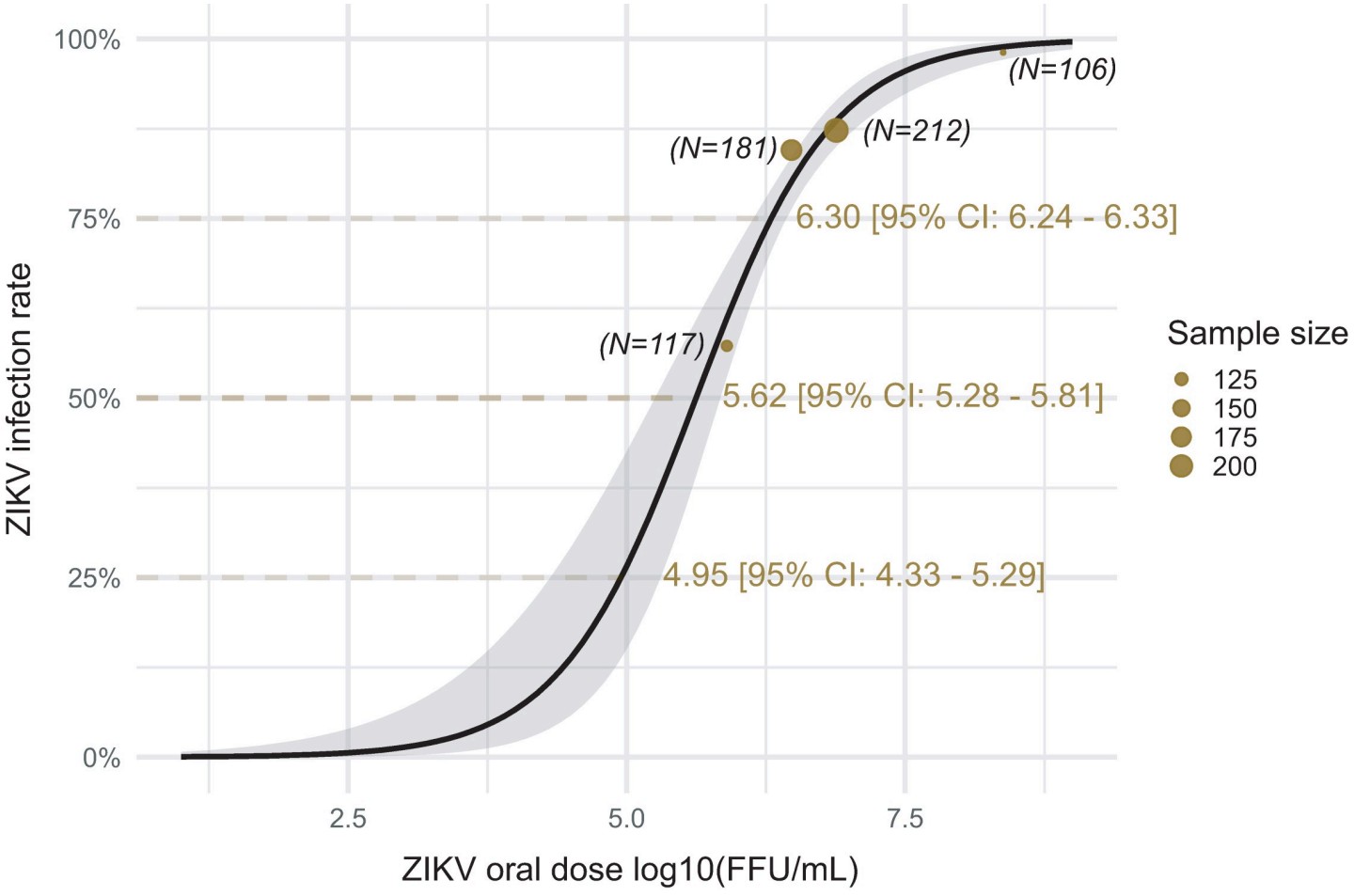

**Fig 3. Dose response curve representing *Ae. albopictus* mosquito infection rate as a function of ZIKV oral dose.** Golden dots correspond to infection rates observed on *Ae. albopictus* mosquitoes at each infectious blood-meal titer. Dots size is proportional to the number of samples. The black line corresponds to fit values obtained by fitting a logistic model to the data. The grey ribbon indicates the 95% confidence intervals. The dose to infect 50% of the ZIKV exposed mosquitoes (ID-50), the 25th and 75th percentile are represented with their 95% confidence intervals.

## Intra-mosquito ZIKV systemic infection dynamics as a function of the oral ZIKV dose

Intra-host dynamics of systemic ZIKV infection was inferred for all virus doses by fitting a logistic model with the virus dose, the time post-exposure and their interaction as explanatory variables. Systemic infection prevalences were significantly influenced by the combination of the time post-exposure and the virus dose (analysis of deviance, interaction term, p-value = 0.03). Systemic infection prevalences saturated at 100% at 25 days post virus exposure for all virus doses but the lowest ($8 \times 10^5$ FFU/mL) for which 96% of mosquitoes had a systemic infection (Fig 4A). The estimated time required to reach 50% of systemic infections was 13.2 days (95% confidence interval: 11.5–15 days), 10.9 days (95% CI: 10–12 days), 9.7 days (95% CI: 9–10.5 days) and 6.8 days (95% CI: 5–8 days) for $8 \times 10^5$ FFU/mL, $3 \times 10^6$ FFU/mL, $7.5 \times 10^6$ FFU/mL and $2.37 \times 10^8$ FFU/mL infectious blood meal concentrations, respectively. Intra-mosquito systemic infection dynamics were inferred from a range of dose from $10^3$ to $10^8$ FFU/mL (Fig 4B). Median time estimates to reach 50% systemic infection were inversely correlated with the virus dose, ranging from 7.34 days to 37 days post virus exposure for the

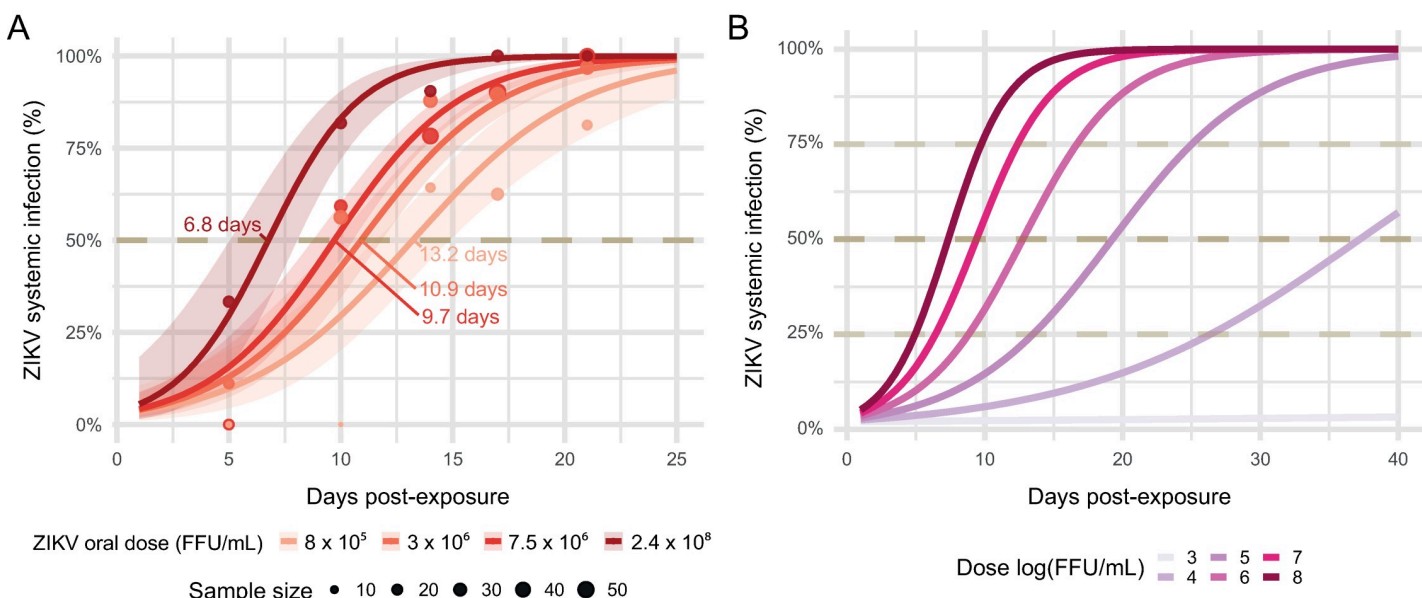

**Fig 4. *Ae. albopictus* mosquito systemic infection rate dynamic post-virus exposure as a function of the oral dose.** (A) Systemic infection rate dynamics observed experimentally. Dots correspond to systemic infection rates observed on *Ae. albopictus* mosquitoes at each infectious blood-meal titer. Dots size is proportional to the number of samples. The time-dependent effect of the virus dose on systemic infection rates was modelled using logistic regression. Lines correspond to fit values with their 95% confidence intervals displayed as ribbons. (B) Predicted systemic infection rate dynamics according to the virus dose and time post-virus exposure for a range of infectious blood meal titers (3 to 8 log10 FFU/mL).

$10^8$ FFU/mL and $10^4$ FFU/mL doses, respectively. The lowest virus dose ($10^3$ FFU/mL) was not sufficient to produce systemic infection in mosquito in the time range from which values were predicted (1–40 days).

## Relationship between intra-human ZIKV load dynamics and intra-mosquito systemic infection rate dynamics

We used our model of intra-human ZIKV load dynamics to generate ZIKV load dynamic variability by running the model on 1,000 independent iterations and sampling for each iteration a parameter value from a normal distribution in the form $N(\mu$ = mean parameter estimate, $\sigma2 = \mu/6)$, with 6 being a scaling factor determined empirically based on the covering of patient data by the distribution of predicted viremia dynamics (black lines, Fig 1). The mean peak ZIKV viremia was $6.8 \times 10^7$ ZIKV RNA copies per mL ($5^{th}$ percentile: $3 \times 10^7$ ZIKV RNA copies per mL, $95^{th}$ percentile: $1.3 \times 10^8$ ZIKV RNA copies per mL) at a mean time of 2.5 days ($5^{th}$ percentile: 2 days, $95^{th}$ percentile: 3 days) after infection, always reached before symptom onset (black lines, Fig 1A). The median time of infectious ZIKV clearance post-infection was 9 days ($25^{th}$ percentile: 8 days, $75^{th}$ percentile: 10 days, time to event analysis). At day 2 post-infection, a mean concentration of $7.3 \times 10^4$ FFU/mL was observed ($25^{th}$ percentile: $1.6 \times 10^4$ FFU/mL, $75^{th}$ percentile: $1.2 \times 10^5$ FFU/mL). The mean virus concentration dropped to 7 FFU/mL ($25^{th}$ percentile: 0.1 FFU/mL, $75^{th}$ percentile: 2.5 FFU/mL) at day 9 post-infection (*i.e.*, the median time of infectious ZIKV clearance post infection).

Mosquito infection probabilities were inferred throughout the dynamics of viremia in humans. Mosquito infection probability started at 0 and reached its peak at day 2 post-infection (22% according to model prediction using mean fit parameter values as determined in Fig 1A and 1B–the golden line in Fig 1C–with a mean 19.4% ($25^{th}$ percentile: 9%, $75^{th}$ percentile: 29%) as calculated from the 1,000 iterations–black lines) to decrease rapidly and reach 0.3% at

day 6 (i.e., time to symptom onset) with a mean of 0.7%, 25th percentile: 0.2%, 75th percentile: 0.86% as calculated from the 1,000 iterations.

Based on our model parameter estimation, the median time to reach systemic infection was the lowest at day 2 post-infection. At the peak of viremia, half of the infected mosquitoes would need 20.7 days (median 21 days, 25th percentile: 19 days, 75th percentile: 30 days, as calculated from the 1000 iterations) to develop a systemic infection in our experimental settings (Fig 1D). The median time to reach systemic infection was inferior to 50 days in only 20 out of the 1,000 iterations at the time of symptom onset in human (day 6 post-infection), and inferior to 30 days in only 3 out of the 1,000 iterations.

## Integration of the intra-host dynamics into outbreak simulations enhances realism to assess factors influencing outbreak initiation probability and scale

We implemented a stochastic agent-based model (ABM) to assess the epidemiological impact of within-host ZIKV dynamicsin dose-dependent manner using the R package nosoi [30]. Starting with one infected human in a population of susceptible hosts (humans and mosquitoes), the model simulates ZIKV transmissions by considering the dynamic nature of within-human virus loads and the associated probability of mosquito infection and virus transmission timeliness (*i.e.*, Extrinsic Incubation Period, EIP). The model was run 100 independent times over 365 days over a range of mean individual mosquito biting rates: 1, 5, 10 and 60 independent mosquitoes biting per person per day.

The risk of ZIKV outbreak initiation varied significantly across mosquito biting rates densities (p-value = 0.0005, Fisher's Exact Test), with a probability of outbreak initiation success of 0%, 4%, 11% and 53% for 1, 5, 10, and 60 mosquito bites per person per day (Fig 5). Intensities of ZIKV outbreaks, as represented by the number of secondary human infections throughout of the epidemic, were also dependent on the intensity of mosquito bite exposure (Fig 5). Large-scale ZIKV outbreaks (*i.e.*, outbreaks yielding more than 100 secondary human infections) occurred in 25% of the simulations at the highest bite intensity level only (60 bites per person per day). Outbreak dynamic was also related to the intensity of mosquito bite exposure. In our model, ZIKV outbreak terminated (i) in absence of infected humans in the synthetic population or (ii) when the maximum number of human or mosquito cases (i.e. 100,000 and 1,000,000 respectively) was reached. The maximum threshold of 1,000,000 mosquito infections was only reached at the highest mosquito biting intensity (*i.e.*, 60 bites per person per day). The distribution of the number of secondary cases directly generated by one infectious case in a susceptible human population is a good indicator of an outbreak dynamic. Secondary cases values distributions across simulations were wide for all conditions, which reflect the stochastic nature of an outbreak. Mean secondary case values increased as a function of the mosquito biting intensity with a mean $0.0 \pm 0.0$ (Mean ± SD), $0.08 \pm 0.4$, $0.30 \pm 0.9$, and $1.1 \pm 1.8$ for 1, 5, 10, and 60 mosquito bites per person per day, respectively. The mean time separating a mosquito infection to the first virus transmission (EIP) was $24.2 \pm 11.1$ (Mean ± SD, calculated with 9 infectious mosquitoes over the 100 run replicates), $17.8 \pm 9.4$ (N = 35 infectious mosquitoes across replicates) and $17.9 \pm 10.2$ days (N = 238,862 mosquitoes across replicates) for the for 5, 10, and 60 mosquito bites per person per day, respectively (there was no infectious mosquitoes for the lowest bite density). These results are coherent with the median time distribution to reach systemic infection at day 2 post-infection calculated previously.

## Discussion

While the mosquito *Aedes hensilli* was strongly suspected of transmitting ZIKV during the Yap island outbreak [9], the yellow fever mosquito *Aedes aegypti* has been incriminated in all

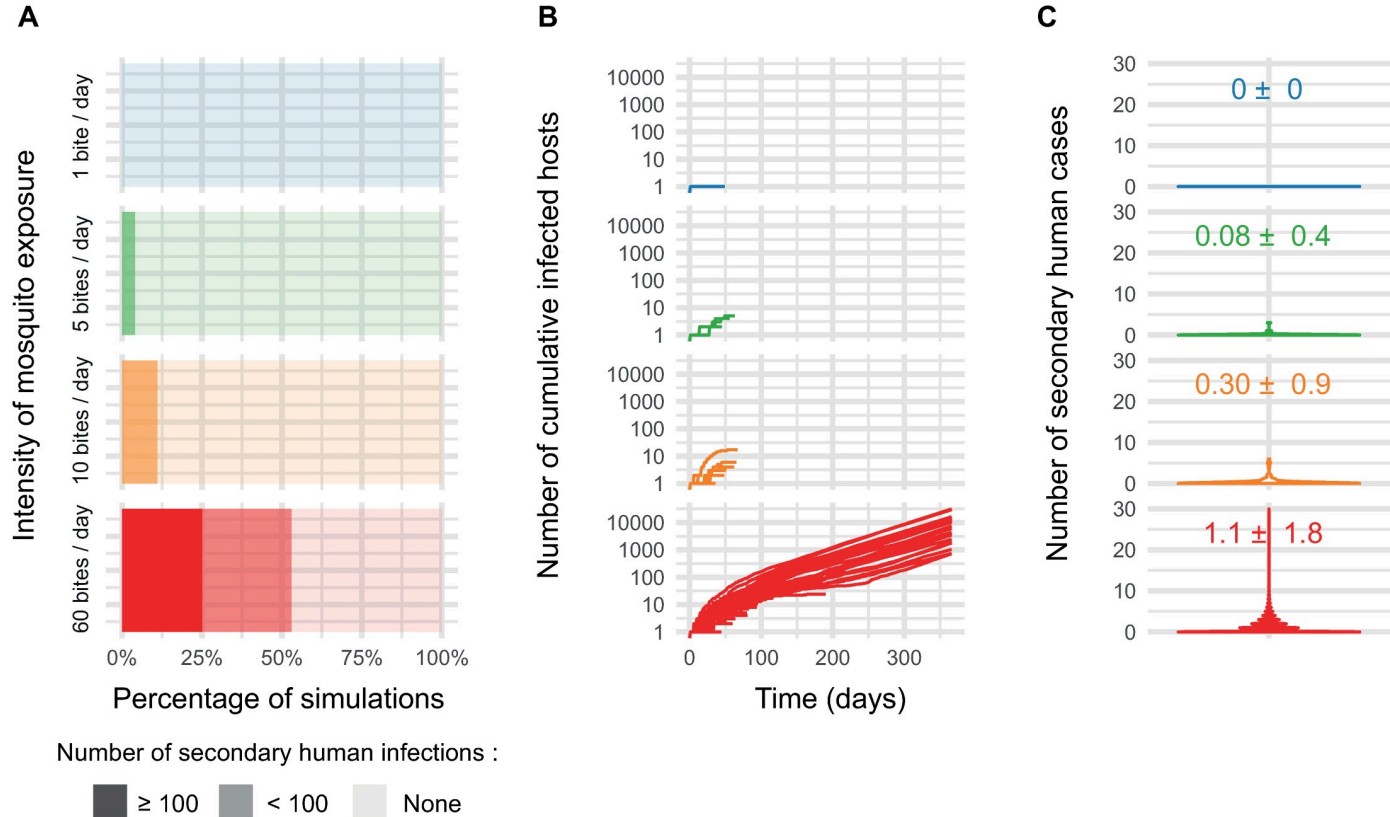

**Fig 5. ZIKV outbreak simulations results according to levels of mosquito bites exposure and the consideration of the dose-dependent intra-mosquito ZIKV dynamic.** Stochastic agent-based epidemiological simulations considering within-host infection dynamics on transmission probability during mosquito-human infectious contacts were performed in 100 independent replicates. A total of 4 mosquito bite intensity levels were tested: 1, 5, 10, and 60 bites per human per day. (A) Stacked proportions of outbreaks resulting in no secondary infected human host, infected human hosts < 100 and infected human hosts ≥ 100. (B) Cumulative number of infected humans over time. Each curve represents a simulation run. (C)- Violin plots showing the number of secondary cases values densities for each condition. Mean number of secondary cases ± Standard Deviation (SD) is represented in each panel.

major outbreaks that have occurred in the Pacific Ocean, South-America or the Caribbean, especially in urban settings [31]. The Asian tiger mosquito *Ae. albopictus* is considered as the second most important vector of human viral pathogens. Originating from Asia, this species has spread throughout the world and is now present in almost all continents, including in tropical and temperate regions. Although considered as having more general blood-feeding preferences than *Ae. aegypti* (the latter has a strict anthropophilic behavior), *Ae. albopictus* is well adapted to the urban environment and can be an aggressive biter and an important nuisance for the human populations [32]. This species was found to be competent for ZIKV experimentally [23,26,31,33–36], and strongly suspected to sustain a ZIKV epidemic activity (African lineage) in Gabon in 2007 [8]. *Ae. albopictus* was also implicated recently in autochthonous ZIKV (Asian lineage) transmissions in the South of France [27], which provides clear confirmation of its vector potential. However, it is not known if *Ae. albopictus* could trigger large scale epidemics in areas where the species exists.

To explore this potential, we studied the dynamic nature of vector competence of *Ae. albopictus* for ZIKV. We assessed the link between ZIKV loads dynamic in humans and *Ae. albopictus* infection probability and systemic infection dynamics. One major originality of our study is to consider the inter-relationship between both within-hosts dynamics involved in the virus life-cycle. We aimed to provide a global and realistic view of *Ae. albopictus* vector

competence for ZIKV by considering vector competence as time and virus dose-dependent. Our study revealed that the mosquito infection rate increases as a function of the infectious blood meal dose, independently of the time post-exposure. Infections rates were shown to be highly variable across studies: a meta-analysis of the literature that focused on *Ae*. albopictus' competence for ZIKV revealed an overall high estimated infection rate of 79% (95% CI 69%–89%) 7 days post-exposure with significant heterogeneity across studies [37]. To our knowledge, only two studies have assessed *Ae. albopictus* infection rates as a function of the ZIKV dose [25,26]. The oral median infection dose (ID50) estimated by these studies ranged from 6.1 to 6.8 $\log_{10}$ FFU/mL. Our ID50 estimate (5.6 $\log_{10}$ FFU/mL (95% CI: 5.3 $\log_{10}$ FFU/mL—5.8 $\log_{10}$ FFU/mL)) was in the same order of magnitude and suggested that the median infection dose in *Ae. albopictus* is not strongly influenced by mosquito populations and/or viral isolates. We revealed that these ID50 values are higher than the maximum ZIKV load estimates occurring in the course of the viremia post-infection in humans (which was estimated to be about 5 $\log_{10}$ FFU/mL). We estimated that the mosquito infection probability was around 20% at the peak of viremia in humans (mean concentration of $7.3 \times 10^4$ FFU/mL at day 2 post-infection). This optimum probability of mosquito infection is very transient, and drops to 0.3% at day 6 post-infection in humans (*i.e*., average time to symptom onset).

Low systemic infection rates for ZIKV and *Ae. albopictus* were often revealed in several independent studies [23,24]. These results are in apparent contradiction with our findings showing levels of systemic infection rates that reach 100% in both Marseille (Metropolitan France) and La Réunion (Indian Ocean) populations from 5 $\log_{10}$ FFU/mL upwards. Similar high systemic infection rates were also described in the literature after 14 days post virus exposure and for ZIKV infectious doses above 5 $\log_{10}$ PFU/mL [25,26,36,38]. It is not straightforward to identify factors underlying these systemic infection rate discrepancies across studies. Mosquito vector competence for viruses was reported to differ according to mosquito populations [39,40], virus isolates [41], their combinations [42], or the virus dose [25,26]. More complex interactions between intrinsic (*e.g*., mosquito virome [43], endogenous non-retroviral elements [44,45], or bacterial microbiome [46]) or environmental (*e.g*., temperature [47,48], rearing or experimental settings) conditions might further impede comparisons across vector competence studies [49]. Systemic infection dynamic was strongly influenced by the infectious blood meal dose, with a 30 days shift of median time estimates to reach 50% systemic infection between 5 $\log_{10}$ FFU/mL and 8 $\log_{10}$ FFU/mL. At the peak of Zika viremia in humans, half of the infected mosquitoes would need around 21 days to develop a systemic infection in our experimental settings. Aside for highlighting the high risk of ZIKV transmission during the pre-sympotmatic stage of the infection, these results revealed a relatively low epidemic potential of *Ae. albopictus* for ZIKV. Tesla B. and colleagues have previously empirically demonstrated the dependency of infection rate, systemic infection and transmission dynamic on ZIKV oral viral dose using the *Ae. aegypti* mosquito model [28]. They estimated at 4.98 $\log_{10}$ Plaque Forming Units (PFU)/mL the infectious dose required to infect 50% of the mosquito population (ID50). By applying our analysis procedure (S1 File) on their raw data, we estimated the time estimates to reach 50% systemic infection to be 18.7, 14 and 8.2 days across their 3 replicates for a 5 $\log_{10}$ PFU/mL oral dose. By comparison, we estimated in *Ae. albopictus* the ID50 at 5.6 $\log_{10}$ FFU/mL and at 19.2 days the median systemic infection estimate for a similar ZIKV oral dose (5 $\log_{10}$ FFU/mL, different titration methods were applied between both studies). These results suggest that infection rates and intra-mosquito virus dynamics are similar across *Ae. aegypti* and *Ae. albopictus* mosquito species, despite the documented role of *Ae. aegypti* as the main vector during major ZIKV outbreaks. An overall low intrinsic ability of both species to transmit ZIKV in experimental settings has been previously reported [23], and other parameters might be at play to allow a sustained virus transmission on the field. It has

been previously demonstrated that an urban, *Aedes aegypti*-borne, epidemic of yellow fever occurred in 1987 in Africa although the mosquito vector was relatively resistant to infection and transmitted the virus inefficiently. It was suggested that a high mosquito density can override a relatively low vector competence to sustain active virus transmissions [50].

Combined with blood feeding behaviours, mosquito bite rate and vector longevity, intra-mosquito dynamic is a powerful contributor to vectorial capacity (VC), a restatement of the basic reproductive rate ($R_0$) of a vector-borne pathogen [51,52]. To approximate VC, we build a realistic vector-borne virus transmission framework throught the implementation of Agent-Based Simulations (ABM) based on the within-hosts dynamic inter-relationship. We revealed that the probability of ZIKV outbreak initiation, dynamics, and scale were strongly dependent on the intensity of mosquito bite exposure. The probability of a large ZIKV outbreak ($\geq$ to 100 secondary human cases) to initiate ranged from 0% at 1 bite per person per day to 25% at 60 bite per person per day. The positive influence of the human biting rate parameter on VC has been well depicted by deterministic models representing VC, such as the Ross-Macdonald model [51,53,54]. In these models, vector competence is treated as a limiting factor (a proportion or probability) fixed in time and not influenced by other parameters. We here reached similar conclusions using a different and independent system that can easily integrate the stochastic and dynamic nature of outbreaks. It thus provides a more realistic scenario where vector competence is treated as a dynamic parameter influenced by the virus dose, as highlighted by our experimental results.

An almost unanimous consensus is prevailing on the existence of a low transmission efficiency (*i.e.*, high transmission barrier) [22–24,26,34,35,55] in the *Ae. albopictus*/ZIKV couple. Variation of the transmission barrier permeability as a function of the virus dose or mosquito population was not addressed in our work. The presence of a transmission barrier would further scale down our predictions concerning the epidemic potential of *Ae. albopictus* for ZIKV but would not impact our main conclusions regarding the dependency of intra-mosquito ZIKV infection and systemic infection dynamic on the virus dose. Alternatively, many other factors that have not been considered here can enhance this potential. For instance, higher ZIKV load and prolonged duration of virus detection were encountered in capillary blood as compared to venous blood [56]. It has also been observed that infection rates in mosquitoes are higher when viremic hosts are used as the blood meal source, as compared to artificial infectious blood feeding [57]. Based on our prediction, any increase in virus dose intake by the mosquito would result in a dramatic rise of the epidemic potential of *Ae. albopictus* for ZIKV. In addition, transmission efficiency was reported to be strongly affected by temperature: median EIP was shown to shift from 9.6 days at 26˚C to 5.1 days at 30˚C for *Ae. aegypti* mosquitoes infected with ZIKV [47]. ZIKV might also have the opportunity to evolve towards shorter intra-mosquito dynamics, as had already occurred in other system [58]. Other ZIKV genotypes, such as those from the African lineage, might also lead to more permissive infection and systemic infection barriers in *Ae. albopictus* [59].

Our conclusions concerning the inter-relationship between within-hosts dynamics also rely on a strong hypothesis of similar Zika viremia dynamics profil in human and non-human primates. While clinical manifestations of arbovirus infections often differ from humans in non-human primates, both model appear to share similar estimates of duration of viremia [60,61]. Modelisation of Zika viremia in humans is challenging due to the paucity of data available. Viral loads can generally only be accessed in symptomatic patients after symptom onsets, and it is not straightforward to date the time of infection through an infectious mosquito bite with accuracy. A large proportion of ZIKV infections are asymptomatic, and virus loads dynamic remains elusive for this category of patients. In addition, we simulated a within-human viremia diversity by considering no-correlation across the several parameters of our model, which

might not truly reflect the reality. We aimed to provide the best viremia model based on the few data that were available, and as such any deviations from our initial hypothesis might impact our predictions. However, any additional information provided by future studies would be easily integrated in our ABM model (provided in S2 File) to update our assessement of *Ae. albopictus* as a vector.

In summary, our results revealed a low epidemic potential of *Ae. albopictus* for ZIKV, independently of its tropical or temperate origin, which might explain the absence of large scale ZIKV epidemics in territories occupied by *Ae. albopictus*. This low epidemic potential however cannot be considered as an obstacle to autochthonous ZIKV transmissions as it can rise under favorable conditions (*e.g.*, high mosquito density) to trigger large scale ZIKV outbreaks.

## Materials and methods

### Ethics statement

Institutional review board approval was granted by the Comité de Protection des Personnes Sud-Méditerranée I corresponding to the following study "Etude descriptive prospective de la maladie à virus Zika au sein de la communauté de défense des Forces Armées en Guyane" and was registered February 2016 under the number RCB: 2016-A00394-47. Written informed consent was obtained from each patient as required by the Comité de Protection des Personnes Sud-Méditerranée I.

### Zika virus viremia dynamic in human

**Data for ZIKV plasma loads in human over time.** Venous serum samples from 35 symptomatic Zika virus patients were collected at different time points post-symptoms onset. All patients were infected during a period ranging from March to September 2016 in French Guiana in the course of the ZIKV outbreak in the American continent starting in 2015. Twenty-one of these patients were included in a prospective study that described ZIKV genomic RNA load in sera samples collected sequentially from venous and skin capillary blood [56]. Thirteen additional patients from whom sera were only collected from venous blood were added in this study (virus loads for these patients can be considered here as primary data) [62]. ZIKV genomic RNA was extracted from 150 μL of each sample using the QIAamp Viral RNA kit (QIAGEN, Hilden, Germany). ZIKV loads, expressed as the number of genomic RNA copies per mL, were determined using the RealStar Zika Virus real-time reverse-transcription PCR (RT-PCR) Kit 1.0 (Altona Diagnostics GmbH, Hamburg, Germany) according to the manufacturer's instructions and a standard curve generated using 10-fold serial dilutions of quantified total ZIKV RNA (28930015). All patients were followed up until the absence of ZIKV RNA detection in their venous serum.

Importantly, we had access to one sample collected one day before symptoms onset. To our knowledge, this represents the only human serum sample that has been collected in the pre-symptomatic stage of the disease so far. Only the quantification cycle (Cq) values could have been recovered for this patient because RT-PCR was performed without a standard curve. One ZIKV load (RNA copies/mL) value was also missing for another patient at his first time-point post-symptoms onset. Because these samples were no longer available at the time of data analyses, and because we consider that they were critical to model ZIKV viremia over time post infection, ZIKV loads (RNA copies/mL) were inferred from Cq values based on a linear regression analysis performed with RT-PCR data from the other samples and diluted standards. While performing absolute quantitation based on comparison of Cq values that were not obtained from the same experiment could lead to inaccurate estimates [63], we deliberately chose this approach because the same RT-PCR procedure was used across experiments

independently of the presence of a standard curve. Patients without detectable virus load in their plasma were discarded from the analysis. A maximum of 4 plasma samples was available per patient. In order to improve parameters estimation in the subsequent model fitting step, a ZIKV plasma load dynamic was obtained post-symptom onsets by averaging ZIKV loads across 23 patients (S3 and S4 Files).

**Fitting a virus load dynamic model to data.** The within-human host plasma viral load dynamic was assessed using a target-cell limited model with a latent phase, as represented by a system of 4 ordinary differential equations described by Best *et al.* [29]:

$$\frac{dT}{dt} = -\beta VT; \; \frac{dI_1}{dt} = \beta VT - kI_1; \; \frac{dI_2}{dt} = kI_1 - \delta I_2; \; \frac{dV}{dt} = pI_2 - cV.$$

This model was used to model plasma Zika virus dynamics in nonhuman primates. $T$, $I_1$, and $I_2$ represent the concentration (cells per mL) of (i) uninfected target cells, (ii) infected target cells that do not produce virus yet, and (iii) productive infected cells, respectively. $V$ represents the concentration of free virus particles (ZIKV RNA copies/mL). We assumed that infection is initiated by the introduction of $V_0$ ZIKV RNA copies into the human host during the mosquito bite. Free viruses ($V$) infect target cells ($T$) at rate $\beta$TV. This constant infection rate of target cells per virion ($\beta$) is a limiting factor affecting the probability of a virion to enter a cell. Infected cells then enter in a latent phase ($I_1$) where they do not produce virus and transition to a virus productive state ($I_2$) at rate $k$ per day, with $1/k$ being the average transition time from $I_1$ to $I_2$. Productively infected cells ($I_2$) release free viruses at a rate $p$ per day and die at a rate $\delta$ per day, where $1/\delta$ is the average lifespan of productively infected cells. Free viruses are cleared at a rate $c$ per day due to the effect of the immune response or virus decay. Target cell replenishment was ignored in the model because of the short-term nature of the infection.

The model was fitted to observed data using a Bayesian approach implemented in Stan [64] through the Rstan package [65] (S3 and S4 Files). A fixed incubation period of 6 days was added to each time post-exposure in our data prior to fit the model in order to standardize the time scale between our data and the model (*i.e.*, our data are expressed in time post-symptom onset while the model start at the day of infection). This six-day incubation period corresponds to the rounded averaged median time of the ZIKV incubation period estimated from different independent studies (i.e. 5.9 days (95% credible interval, 4.4–7.6 days) for Lessler J. *et al.*, 6.2 (95% CI 5.7––6.6) for Krow-Lucal E.R. *et al.*, and 6.8 (95% CI, 5.8–7.7 days) for Fourié T. *et al.*) [66–68]. The lack of data before symptom onset precludes the estimation of a unique combination of model parameter values that could fit the data with a family of solutions. To address this issue, we used parameter estimates inferred from non-human primate viremia dynamics as strong priors in our model fitting process under the strong hypothesis that human and non-human viremia dynamics are closely related [69]. Prior distributions were adapted from Best *et al.* [29]. The initial virus load distribution was modified to be into the range of ZIKV titers found in *Aedes aegypti* saliva [70] (around 100 infectious viruses for an estimated volume of 10 nL of saliva [71]). The initial normal prior distributions for virus load and target cells concentration, denoted *V0* and *T0* respectively, were set to *V0*: $N(\mu = 10{,}000, \sigma2 = 500)$ and *T0*: $N(\mu = 1e5, \sigma2 = 1e4)$, respectively. The initial concentration of infected cells, $I_1$ and $I_2$, was assumed to be zero. Parameters $\beta$, $k$, $\delta$, $p$ and $c$ were sampled from the respective normal prior distributions $\beta$: $N(\mu = 2e\text{-}7, \sigma2 = 1)$, $k$: $N(\mu = 6, \sigma2 = 3)$, $\delta$: $N(\mu = 2, \sigma2 = 1)$, $p$: $N(\mu = 25{,}000, \sigma2 = 2{,}500)$ and $c$: $N(\mu = 10, \sigma2 = 3)$. The model was run on four independent chains over 2,000 iterations with sampling beginning after 1,000 warmup iterations. Parameter estimates have an effective sample size $> 1{,}000$ and a potential scale reduction factor (*R-hat*) of 1, both of which ensure convergence and a lack of autocorrelation. Prior posterior overlaps

(PPO) were calculated with the MCMCvis R package [72]. Parameters with a PPO >35% were not considered robustly identifiable [73].

Predicted ZIKV RNA loads (copies/mL) were converted into infectious virus loads (FFU/mL) by dividing the viremia predicted by the model by a fixed arbitrary value of 750. This value lies at the midrange of the RNA copies/PFU ratios (infectious titers were reported to be 500–1,000-fold less than ZIKV RNA copies concentrations) that were estimated in non-human primate plasma with a French Polynesian ZIKV isolate [74].

## Zika virus isolate

In this study, we used the ZIKV isolate SL1602 that was isolated from the plasma of a traveler returning from Suriname in 2016 [75]. This isolate belongs to the Asian lineage and is phylogenetically close to other viruses that have recently been isolated in the Americas [75]. The virus was passaged four times in C6/36 cells (*Aedes albopictus*) prior to its use for mosquito infections. To prepare virus stock, sub-confluent C6/36 cells were infected using a viral multiplicity of infection (MOI) of 0.1 (one infectious viral particle for 10 cells) in 25-cm$^2$ culture flasks and incubated for 7 days at 28˚C with 5 mL of Leibovitz's L-15 medium supplemented with 0.1% penicillin (10,000 U/mL)/streptomycin (10,000 μg/mL) (Life Technologies, Grand Island, NY, USA), 1X non-essential amino acids (Life Technologies) and 2% fetal bovine serum (FBS, Life Technologies). At the end of the incubation, the cell culture medium was harvested, adjusted to 10% FBS, and pH ~8 with sodium bicarbonate, aliquoted, and stored at −80˚C. An additional 5 mL of Leibovitz's L-15 medium prepared as described above were added to infected confluent C6/36 cells and harvested 3 days later. This procedure increases virus titres in the stock solution.

Virus titration was performed by focus-forming assay (FFA) on one aliquot that has been stored at −80˚C, as previously described with minor modifications [76]. This assay relies on inoculation of 10-fold dilutions of a sample onto a sub-confluent culture of C6/36 cells, followed by incubation and subsequent visualization of infectious foci by indirect immunofluorescence. Modifications to the previously published protocol include a 1-hour incubation step at 37˚C with 40 μL/well of mouse anti-*Flavivirus* group antigen antibody clone D1-4G2-4-15 (Merck Millipore, Molsheim, France) diluted 1:200 in PBS + 1% bovine serum albumin (BSA) (Interchim, Montluçon, France). After another three washes in PBS, cells were incubated at 37˚C for 30 min with 40 μL/well of a goat anti-mouse IgG (H+L), FITC-conjugated antibody (Merck Millipore). After three more washes in PBS and a final wash in ultrapure water, infectious foci were counted under a fluorescent microscope and converted into focus-forming units/mL (FFU/mL). The titre of the frozen virus stock was estimated at $6.35 \times 10^5$ focus-forming units per mL (FFU/mL) for the first medium culture harvest and $2.75 \times 10^8$ FFU/mL for the second medium culture harvest. All infectious experiments were conducted in a Biosafety Level-3 (BSL-3) insectary (IHU Méditerranée Infection, Marseille).

## Mosquito populations

Two mosquito populations of *Aedes albopictus* species were used in this study, one from Marseille (Metropolitan France) and the other from La Réunion island, a French overseas territory in the Indian Ocean. *Ae. albopictus* mosquitoes from Marseille were collected both as eggs and adults using ovitraps and human landing catches in three different locations in Marseille city (5th, 12th, and 13th districts) in June 2018. *Ae. albopictus* mosquitoes from La Réunion were all collected as eggs in June 2018 from six different localities named Sainte Marie Duparc, Saint André centre-ville, Saint Paul, Le Port Rivière des Galets, Saint Leu Pointe des châteaux, Saint Pierre Bois d'olive (Fig 6). A first-generation population was obtained from the eggs of adults

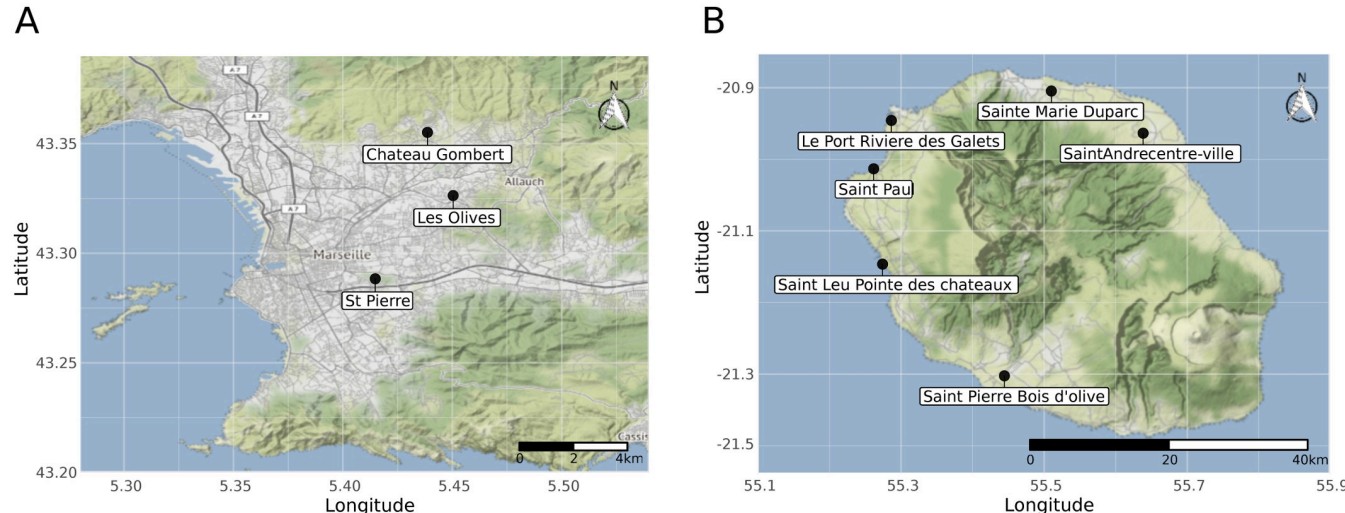

**Fig 6. Geographic location of the F0 mosquito collecting sites in Marseille (A) and Réunion island (B).** Mosquitoes from these sites were the wild progenitors of the Marseille (Metropolitan France) and La Réunion (Overseas France, Indian ocean) populations (4[th] generation) used in this study. Source: Base map and data from OpenStreetMap and OpenStreetMap Foundation.

originating from ovitraps or captured by human landing catches. For each population, larvae from different collection sites were combined, and adult mosquitoes were maintained in standardized insectary conditions (28°C, 75 ± 5% relative humidity, 16:8 hours light-dark cycle) until the 4[th] generation by mass sib-mating and collective oviposition. Adult females from each population were blood-fed on human blood obtained from the Etablissement Français du Sang (EFS) through a membrane feeding system (Hemotek Ltd, Blackburn, UK) using pig intestine as the membrane. Access to human blood was based upon an agreement with the EFS. Eggs were hatched in reverse osmosis water, and larvae were reared with a standardized diet of fish food in 24 × 34 × 9 cm plastic trays at a density of about 400 larvae per tray. A maximum of 800 male and female adults was maintained in 24 × 24 × 24 cm screened cages with permanent access to a 10% sucrose solution.

## Experimental mosquito infections

Nine to 13-day old females were transferred to the BSL-3 and deprived of sucrose solution 24 hours before experimental virus exposure. Sixty to 80 females were confined into 80-mm high and 80/84 mm (inside/outside) diameter cardboard containers. Containers were sealed on the top with mosquito mesh and with a 65-mm high polystyrene piston covered up with a plasticized fabric that matches the inside diameter of the container at the bottom. Pistons of each container were rose before to infectious blood meal exposure to contain all mosquitoes in a tight space below the infectious blood meal. This procedure increases the yield of engorged mosquitoes. Females were allowed to feed for 20 min from an artificial feeding system (Hemotek) covered by a pig intestine membrane that contained the infectious blood meal maintained at 37°C. Feeders were placed on top of the mesh that sealed the containers. The infectious blood meal consisted of two volumes of washed human erythrocytes and one volume of viral suspension. Human erythrocytes were collected and washed one day before the experimental infection. An aliquot of the artificial blood meal was collected immediately before blood-feeding and titred by FFA, as described above but without a freezing step.

Experimental mosquito exposure to ZIKV was performed in three independent experiments. Mosquitoes from both Marseille and La Réunion populations were exposed to the same

infectious blood meal in experiments 1 and 2. In experiment 3, mosquitoes from the La Réunion population only were exposed to two different infectious blood meal titers. Final ZIKV virus titres in blood meals were $7.5 \times 10^6$ FFU/mL and $3 \times 10^6$ FFU/mL for experiment 1 and 2, respectively and $2.37 \times 10^8$ FFU/mL and $8 \times 10^5$ FFU/mL for experiment 3. After virus exposure, fully engorged females were cold anesthetized and sorted on ice before being individually transferred to new cardboard containers and maintained under controlled conditions (28°C, 75 ± 5% relative humidity, 16:8 hours light-dark cycle with permanent access to a 10% sucrose solution).

## ZIKV RNA detection

ZIKV RNA was detected in mosquito bodies and heads after a fresh dissection from freeze-killed mosquitoes at 5,10, 14, 17, and 21 days post virus exposure (DPE) for both *Ae. albopictus* populations. The presence of ZIKV RNA in bodies indicates a midgut infection, while the presence of the virus RNA in mosquito heads indicates a systemic (disseminated) infection [41]. These two vector competence indices were determined qualitatively (i.e., presence or absence of virus in mosquito bodies and heads, respectively). Dissected mosquito heads and bodies were homogenized individually in 400 μL of lysis buffer (NucleoSpin 96 Virus Core Kit, Macherey-Nagel, Hoerdt, France) during three rounds of 20 sec at 5,000 rpm in a Precellys 24 mixer mill (Ozyme, Saint-Quentin-en-Yvelines, France). Viral RNA from individual organs was extracted using the NucleoSpin 96 Virus Core Kit (Macherey-Nagel) according to the manufacturer's instructions. At the final step, viral RNA from each sample was eluted in 100 μL of RNase-free elution buffer.

Detection of ZIKV RNA was performed with an end-point reverse transcription polymerase chain reaction (RT-PCR) assay. ZIKV genomic RNA was first reverse transcribed to complementary DNA (cDNA) with random hexamers using M-MLV Reverse Transcriptase (Life Technologies) according to the manufacturer's instructions. cDNA was amplified by 35 cycles of PCR using the set of primers targeting the RNA-dependent RNA polymerase NS5 gene: ZIKV-F: 5'-GCCATCTGGTATATGTGG-3' and ZIKV-R: 5'-CAAGACCAAAGGGGGAG CGGA-3'. Amplicons (393 bp) were visualized by electrophoresis on 1.5% agarose gels.

## Statistical analysis

The time-dependent effect of the mosquito population on mosquito body infection and systemic infection was analysed by logistic regression by considering each phenotype as a binary response variable. A full-factorial generalized linear model that included the time post-virus exposure and the mosquito population or the virus dose, depending on the analyses, was fitted to the data with a binomial error structure and a logit link function. Statistical significances of the predictors' effects were assessed by comparing nested models for their changes in deviances (goodness of fit measure for a model) based on a chi-squared distribution. Mosquito populations could be compared for a single dose without the confounding effect of the experiment in experiments 1 and 2. The effect of the virus dose was however confounded with the experiment effect when analysing the virus dose effect on intra-mosquito infection and systemic infection dynamics.

Probabilities of infection or systemic infection according to a virus dose or/and time post virus exposure were estimated with a logistic model in the mathematical form: $\log\left(\frac{P}{1-P}\right) = \beta_0 + \beta_1 \times X1 + \beta_2 \times X2 + \beta_3 \times X1 \times X2 + \varepsilon$, where P denotes the phenotype probability, $\beta_0$ the Y-intercept when all model coefficients equal 0, $\varepsilon$ the error term, and $X_i$ and $\beta_i$ the explanatory variables with their associated coefficients. Infection probabilities and median systemic infection doses were calculated based on logistic regression parameter

estimates (S4 and S5 Files). Time to event analysis was performed using the packages survival [77] and survminer [78]. Fisher's Exact Test with simulated p-value (based on 2000 replicates) was performed to assess the significance of ZIKV outbreak initiation success across biting rates densities. All statistical analyses were performed in the statistical environment R [79]. Figures were made using the package ggplot2 [80] and the Tidyverse environment [81].

## Epidemiological modeling

**Model overview.** To explore how variation in human viremia dynamics can impact the epidemic potential of ZIKV, we performed a series of stochastic agent-based epidemiological simulations using the R package nosoi [82], similar to the simulations performed in Fontaine et al. [41], as a specific branch available on nosoi's GitHub page (https://github.com/slequime/nosoi/tree/fontaine) (S2 File). This framework allows us to take into account within-host infection dynamics on transmission probability during mosquito-to-human and human-to-mosquito infectious contacts and place it in its full epidemiological context. We assumed that transmission could only happen between either an infected human and an uninfected mosquito or between an infected mosquito and an uninfected human. We thus ignored both sexual and vertical transmission as their epidemiological importance is unclear in both mosquitoes and humans, as well as excluded potential superinfections. We also assumed no particular structure in the host or mosquito population. It was assumed that human agents do not die from the infection and leave the simulation when they cure their infection after 12 days. Mosquitoes had a daily probability to die empirically set at 0.15 as used previously [41,83].

## Human-to-mosquito transmissions

Each human agent was exposed to a number of biting load drawn from a Poisson distribution with a mean value empirically set to 1, 5, 10 or 60 individual mosquito bites per person per day. Sixty bites per person per day is relevant to outdoor *Ae. albopictus* females biting activity on La Réunion island [84].

The target-cell limited model with a latent phase was used to simulate Zika viremia in our synthetic environment. For each individual, parameter values were drawn from normal distributions or truncated normal distributions (to avoid negative values) centered on mean parameter estimates ($\mu$, refer to the "*Zika virus viremia dynamic in human*" section of the Materials and Methods) with standard deviations ($\sigma$) equal to mean parameter estimates divided by 6, with 6 being a scaling factor determined empirically based on the covering of patient data by the distribution of predicted viremia dynamics. ZIKV RNA loads (copies/mL) were converted into infectious virus loads (FFU/mL) by dividing by a value drawn from a uniform distribution ranging from 500 to 1,000, based on results showing that infectious titres were 500–1,000-fold less than ZIKV RNA copies concentrations in non-human primate plasma [74]. Mosquito infection probability was inferred for each mosquito-infected human contacts based on the simulated infectious virus dose at the time post-human infection. The probability of human-to-mosquito transmission was inferred from the experimental exposure of *Ae. albopictus* to ZIKV using our logistic model predictions.

## Mosquito-to-human transmissions

A gonotrophic cycle duration, separating two blood-meals, was drawn for each mosquito in a Poisson distribution with a mean of 4 days [85]. Mosquitoes were only allowed to bite one new human agent at times corresponding to multiples of their gonotrophic cycles durations.

Mosquito transmission probability were inferred for each contact from our full-factorial logistic model with the simulated infectious virus dose that initiated the infection in the

mosquito and time post-infection as parameters. Time to systemic infection was used as a proxy for time to transmission (*i.e.*, the extrinsic incubation period, referred to as EIP). We assumed a lack of salivary gland escape barrier and that the virus presence in mosquito head tissues indicated the potential for onward transmission. For each mosquito, an individual EIP value acts as the threshold for active transmission (when time post-infection is greater or equal to this EIP value, then the mosquito can transmit the infection). This individual EIP value was inferred from our time- and dose-dependent logistic regression based on this equation:

$$\text{DD50} = (\log\left(\frac{-P}{P - 1.001}\right) - \beta_0 - \beta_1 \times X1)/(\beta_2 + \beta_3 \times X1)$$

where P = 0.5 (i.e., the median systemic dissemination probability), $\beta_0$ is the Y-intercept value (4.17483), $\beta_1$ (0.10116), $\beta_2$ (-0.25564) and $\beta_3$ (0.08924) are model coefficients associated to the virus dose, time post virus exposure and their interaction, respectively. X1 represents the virus dose value.

**Implementation.** These simulations were run in R version 3.6.4 using the packages foreach [86], doParallel [87], and nosoi [82]; complete R script used is available as S5 File. Briefly, simulations started with one infected human and were run for 365 days or until the allowed number of infected individuals was reached (100,000 humans or 1,000,000 mosquitoes respectively). Each condition was run in 100 independent replicate simulations.

## Supporting information

**S1 Fig. Markov chain Monte Carlo (MCMC) diagnostic plots for the ZIKV load dynamic model fitting.** Trace plots of Markov chain Monte Carlo (MCMC) output for each of the 7 model parameters, with T0: the initial uninfected target cells concentration (cells per mL), V0: initial concentration of free virus particles (ZIKV RNA copies/mL), $\beta$: the infection rate constant of target cells per virion (per mL per day), $k$: rate of cell transition from a non-productive sate to a virus productive state (per day), $\delta$: dying rate of productively infected cells (per day), $p$: free virus releasing rate by productively infected cells (per day) and $c$: virus clearance rate (per day). The model was run on 4 independent chains. The Rhat statistic, the number of effective samples, and the prior posterior overlap (PPO in %) are represented for each parameter. (TIF)

**S2 Fig. Body infection prevalence of *Ae. albopictus* mosquitoes, according to ZIKV titers in the infectious blood meal.** Percentages of body infections over time post-ZIKV exposure are represented at 4 infectious blood meal titer: $8 \times 10^5$ FFU/mL, $3 \times 10^6$ FFU/mL, $7.5 \times 10^6$ FFU/mL and $2.4 \times 10^8$ FFU/mL. 95% confidence intervals are indicated with the error bars, and prevalences averaged across time points are represented with a dashed line for each virus dose. (TIF)

**S1 File. R code related to the intra-mosquito ZIKV dynamic analysis.** The file provides code lines used to analyze the experimental data and perform data visualization and takes S5 File as an input file. This is a an R Markdown file created using RStudio, an open source Integrated Development Environment (IDE) for the R programming language. It contains YAML metadata, markdown-formatted plain text, and chunks of R code that can be rendered using RStudio. (RMD)

**S2 File. R code used to perform the in silico epidemiological simulations with the nosoi R package.** (R)

**S3 File. Database with patient identification number, the declared dates of symptoms, and blood sampling from the 35 patients infected with ZIKV included in our study.** Quantification cycle (Cq) values for each sample and the corresponding ZIKV RNA concentration in copy/mL is provided when available.
(TXT)

**S4 File. R code related to the intra-human ZIKV dynamic analysis.** The file provides code lines used to analyze the experimental data and perform data visualization. This is a an R Markdown file created using RStudio, an open source Integrated Development Environment (IDE) for the R programming language. It contains YAML metadata, markdown-formatted plain text, and chunks of R code that can be rendered using RStudio.
(RMD)

**S5 File. Raw vector competence data.** Each line represents an individual mosquito and the columns are the experimental conditions, and phenotypes. From left to right, the columns indicate the mosquito identification number, the mosquito population (M = Marseille, LR = La Réunion island), number of days post-exposure to the infectious blood meal (dpe), the infectious blood meal concentration (FFU/mL), the body infection status, the head infection status (1 = presence of virus), and the experiment.
(TXT)

## Acknowledgments

We thank Nathalie Wurtz, Muriel Militello, and Jean-Marc Feuerstein for their help and contributions concerning the development of the biological safety level 3 procedure settings concerning experimental mosquito exposure to ZIKV. We also thank Joël Mosnier and Isabelle Fonta for their help with the setting up of BSL3 experiments. We thank Dr. Louis Lambrechts for critically reading an earlier version of this manuscript and providing insightful comments. At last, we thank FJCC for its help with the Stan code on the Stan forum.

## Author Contributions

**Conceptualization:** Sebastian Lequime, Sébastien Briolant, Albin Fontaine.

**Formal analysis:** Sebastian Lequime, Albin Fontaine.

**Funding acquisition:** Albin Fontaine.

**Investigation:** Albin Fontaine.

**Methodology:** Sebastian Lequime, Albin Fontaine.

**Resources:** Jean-Sébastien Dehecq, Séverine Matheus, Franck de Laval, Lionel Almeras.

**Software:** Sebastian Lequime.

**Supervision:** Albin Fontaine.

**Visualization:** Albin Fontaine.

**Writing – original draft:** Sebastian Lequime, Jean-Sébastien Dehecq, Lionel Almeras, Sébastien Briolant, Albin Fontaine.

**Writing – review & editing:** Sebastian Lequime, Albin Fontaine.

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
