## [Decision Letter · Decision Letter 0]

10 Jul 2020

Dear Mr. Fontaine,

Thank you very much for submitting your manuscript "Modeling intra-hosts dynamics of Zika virus transmission reveals the low epidemic potential of Aedes albopictus." for consideration at PLOS Pathogens. As with all papers reviewed by the journal, your manuscript was reviewed by members of the editorial board and by several independent reviewers. In light of the reviews (below this email), we would like to invite the resubmission of a significantly-revised version that takes into account the reviewers' comments.

We cannot make any decision about publication until we have seen the revised manuscript and your response to the reviewers' comments. Your revised manuscript is also likely to be sent to reviewers for further evaluation.

Sincerely,

Elizabeth Ann McGraw, PhD

Associate Editor

PLOS Pathogens

Sonja Best

Section Editor

PLOS Pathogens

Kasturi Haldar

Editor-in-Chief

PLOS Pathogens

orcid.org/0000-0001-5065-158X

Michael Malim

Editor-in-Chief

PLOS Pathogens

orcid.org/0000-0002-7699-2064

Reviewer's Responses to Questions

**Part I - Summary**

Reviewer #1: Here, the authors have produced a paper investigating the likelihood of Aedes albopictus to drive Zika virus epidemics in areas without Aedes aegypti. The authors used experimental vector competence assays and model simulations to estimate the overall vectorial capacity of two populations of Aedes albopictus—1 from Marseilles and 1 from La Reunion. In addition, they were able to estimate the range of viremia in human patients from French Guiana. From these experimental data they were able to parameterize models to better understand the basic reproductive number of these two mosquito populations. Overall, they conclude that Aedes albopictus have a low epidemic potential for Zika virus. While these results are interesting, I believe there are several areas that need to be addressed to facilitate understanding by the reader and a few scientific issues that need to be addressed.

Reviewer #2: This manuscript investigates the vector competence of Aedes albopictus for Zika virus in the context of autochthonous transmission in Europe in 2019. The study combines within-vertebrate and within-mosquito data to develop a model of potential transmission of Zika virus in France. The inclusion of human viral load (especially pre-symptomatic) in humans and the incorporation of this data into the dose-dependence of vector competence was a very interesting and novel piece of the study; but I am confused as to whether this represents primary data rather than a use of already published data (see Major Issues). However, the study does have a novel inclusion of the within-host dynamics of ZIKV and is well developed. There remain some concerns.

**Part II – Major Issues: Key Experiments Required for Acceptance**

Reviewer #1: Critically, others have reported that Aedes albopictus are more susceptible to ZIKV infection than Aedes aegypti, but have a reduced transmission capacity, indicative of a transmission barrier. Unfortunately, since saliva was not collected and screened here, it is not possible to understand if this trend holds true with geographically distinct populations of Aedes albopictus. As a result, it is inaccurate throughout to say that vector competence was measured because transmission potential was not assessed. This should be explicitly stated along with the limitations.

Line 114-116: vector competence, as a component of vectorial capacity, is governed by intrinsic factors that influence the ability of a mosquito species to vector a pathogen and is not interchangeable with the concept of vectorial capacity. Therefore, the time post-exposure when a mosquito becomes infectious—or the extrinsic incubation period—is a key factor that influences vectorial capacity, not vector competence.

Line 131: I believe that it would be useful to present the data for the estimate/establishment of physiologic human bloodmeal titers prior to describing any mosquito infection results. This is critical for understanding the physiologic relevance of your experimental vector competence assays.

Line 178: This section should explicitly state the temperature exposed mosquitoes were held at, because it has recently been reported that increased temperatures reduce ZIKV transmission rates in Aedes albopictus (PMID:31894724). Is 28C reflective of the climate in Marseilles and La Reunion?

Line 256: How is mosquito population density factored into the model described here? And if population density was factored in, is it representative of the overall population density of Aedes albopictus present in Marseille and/or La Reunion?

Line 263: How was the EIP parameterized? This was not measured in your experimental system. It also is not clear how the daily probability of a host being fed upon and daily survival were factored in or estimated. Overall, using these data to estimate the overall vectorial capacity of the two populations of mosquitoes would be worthwhile.

Line 345-348: here it appears that you are arguing that Aedes albopictus has a low epidemic potential because it might take upwards of 21 days for them to transmit virus. There are numerous examples of vector-pathogen relationships for which old-aged mosquitoes are the drivers of transmission and can successfully sustain epidemics. It would be worthwhile to discuss other factors that might make Aedes albopictus less likely to drive epidemics compared to Aedes aegypti, for example. In a previous section you allude to the fact that Aedes albopictus is more catholic in their feeding preferences. This could have an important role in their ability to maintain epidemic transmission.

Line 365: low compared to what? There are several examples of mosquitoes with low competence but high population density that have been capable of sustaining arbovirus outbreaks, e.g. YFV.

Line 379-393: given all of the caveats associated with ZIKV viremia in humans, would it be possible to, for example, use data on dengue virus—for which there is an extensive literature on viremia in humans—to demonstrate the accuracy of your predictions?

Reviewer #2: Methods regarding human viral loads: It is not immediately clear how ZIKV was quantified using RT-PCR. As described (lines 561-581), there is no probe or lightcycler so how is a Cq value derived? Was the human data already published, this was not very clear as the methods makes it seem as this information is primary data.

Line 487: Studies have demonstrated that serial passage in C6/36 cells can attenuate the virus in mosquitoes. Was a comparison made to virus passaged at least once through a mammalian cell line?

Line 536: 9-13 days post emergence is quite old for a vector competence study. What is the justification for this?

Overall: The focus of the paper seems to be more about the dose-dependence leading to this within-vector heterogeneity. The title and conclusions could be better stated to match what the experimental design and structure of the results suggest is the premise of the study.

**Part III – Minor Issues: Editorial and Data Presentation Modifications**

Reviewer #1: Line 34-38 are inaccurate. Aedes albopictus was implicated as the vector of a Zika virus outbreak in Gabon in 2007: PMID: 24516683.

Line 77-80: The statements here are confusing. There are other mosquito species that vector pathogens of public health concern.

Line 81: “continents” should be “inhabited continents”—I don’t believe Aedes albopictus is present on Antarctica.

Line 93: Given all of the emerging evidence, it is inaccurate to characterize ZIKV infection as “usually mild”.

Line 102-104: The references cited do not directly compare ZIKV infection, dissemination, and transmission rates in Aedes aegypti and/or Aedes albopictus to those of the same mosquitoes exposed to DENV, CHIKV, or YFV (and one is a meta analysis), so is therefore a misrepresentation of the available data.

Line 134: please list the geographic origin of the ZIKV isolate used.

Line 142-143: what was the time point for these infection prevalence results?

Line 195: this section should be presented first in the Results.

Line 213-214: How did you establish the PFU:particle ratio here?

Line 310-312: other previous studies have considered the inter-relationship between both within host dynamics.

Line 314: Other studies have demonstrated a dose-dependent relationship between bloodmeal titer and Aedes albopictus infection rates. Those should be discussed and your study placed in context with what has been demonstrated previously.

Line 356: Aedes albopictus is notorious for taking multiple, intermittent bloodmeals, and bloodmeals while still carrying eggs, so a daily bite rate of 1 may not be biologically accurate and higher bite rates may be more in line with what happens in nature. In sum, it is not clear to this reviewer whether this was factored into this statement in the manuscript.

Line 484: was your ZIKV isolate used in your vector competence studies sequence confirmed and what if any reported differences were there between it and the GenBank sequence?

Figure 1: It is difficult to distinguish the size of the dots that represent sample size. It also is not immediately clear what the difference is between light and dark red and blue. And I confess it took me a few minutes of searching to realize the timepoint was listed only in panels C and D.

Figure 2: There are only 3 dots on the graph but 4 dots in the legend. I suggest making the legend reflect what is shown on the graph.

Reviewer #2: Please indicate the type of model used in the abstract.

Reference needed for sentence at Lines 114-116 (The intrinsic ability of…)

Line 464: What is the range of these averaged medians? Model sensitivity to incubation periods has been demonstrated, and understanding the range of this parameter is important for inferring model robustness.

Line 645: References needed for systemic infection as a proxy for transmission since detection in saliva has become the norm for transmission estimation. This also needs to be clearly stated as a caveat of the paper and model in the Discussion.

PLOS authors have the option to publish the peer review history of their article (what does this mean?). If published, this will include your full peer review and any attached files.

Reviewer #1: No

Reviewer #2: No
---

## [Decision Letter · Decision Letter 1]

14 Oct 2020

Dear Mr. Fontaine,

We are pleased to inform you that your manuscript 'Modeling intra-mosquito dynamics of Zika virus and its dose-dependence confirms the low epidemic potential of Aedes albopictus' has been provisionally accepted for publication in PLOS Pathogens.

Best regards,

Elizabeth Ann McGraw, PhD

Associate Editor

PLOS Pathogens

Sonja Best

Section Editor

PLOS Pathogens

Kasturi Haldar

Editor-in-Chief

PLOS Pathogens

orcid.org/0000-0001-5065-158X

Michael Malim

Editor-in-Chief

PLOS Pathogens

orcid.org/0000-0002-7699-2064

Reviewer Comments (if any, and for reference):

Reviewer's Responses to Questions

**Part I - Summary**

Reviewer #1: The authors made a majority of the recommended changes requested during initial peer-review of this manuscript and if the changes were not made, a sufficient explanation was provided. The changes significantly enhanced the credibility and scientific nature of the manuscript. The readers of the article can now fully understand the scientific methods used throughout this study and accurately interpret the scientific findings without bias or incomplete information. I recommend that this article should be accepted for publication without additional major changes to the manuscript.

My only additional comment is to include a statement along the lines of "the virus stock used for vector competence experiments was received from EVAg but not further authenticated".

Reviewer #2: All concerns have been addressed sufficiently

**Part II – Major Issues: Key Experiments Required for Acceptance**

Reviewer #1: (No Response)

Reviewer #2: (No Response)

**Part III – Minor Issues: Editorial and Data Presentation Modifications**

Reviewer #1: (No Response)

Reviewer #2: (No Response)

PLOS authors have the option to publish the peer review history of their article (what does this mean?). If published, this will include your full peer review and any attached files.

Reviewer #1: No

Reviewer #2: No

---

## [Editor Report · Acceptance letter]

4 Dec 2020

Dear Mr. Fontaine,

We are delighted to inform you that your manuscript, "Modeling intra-mosquito dynamics of Zika virus and its dose-dependence confirms the low epidemic potential of Aedes albopictus," has been formally accepted for publication in PLOS Pathogens.

Best regards,

Kasturi Haldar

Editor-in-Chief

PLOS Pathogens

orcid.org/0000-0001-5065-158X

Michael Malim

Editor-in-Chief

PLOS Pathogens

orcid.org/0000-0002-7699-2064